# Identification of Structural Determinants of the Transport of the Dehydroascorbic Acid Mediated by Glucose Transport GLUT1

**DOI:** 10.3390/molecules28020521

**Published:** 2023-01-05

**Authors:** Marcelo Villagrán, Carlos F. Burgos, Coralia I. Rivas, Lorena Mardones

**Affiliations:** 1Research Laboratory in Biological Sciences, Department of Basic Sciences, Medicine Faculty, Universidad Católica de la Santísima Concepción, Concepción 4090541, Chile; 2Laboratory of Neurophysiology, Department of Physiology, Faculty of Biological Sciences, Universidad de Concepción, Concepción 4030000, Chile; 3Laboratory of Antioxidants, Department of Pathophysiology, Faculty of Biological Sciences, Universidad de Concepción, Concepción 4070386, Chile

**Keywords:** GLUT1, glucose transporter, dehydroascorbic acid

## Abstract

GLUT1 is a facilitative glucose transporter that can transport oxidized vitamin C (i.e., dehydroascorbic acid) and complements the action of reduced vitamin C transporters. To identify the residues involved in human GLUT1’s transport of dehydroascorbic acid, we performed docking studies in the 5 Å grid of the glucose-binding cavity of GLUT1. The interactions of the bicyclic hemiacetal form of dehydroascorbic acid with GLUT1 through hydrogen bonds with the -OH group of C3 and C5 were less favorable than the interactions with the sugars transported by GLUT1. The eight most relevant residues in such interactions (i.e., F26, Q161, I164, Q282, Y292, and W412) were mutated to alanine to perform functional studies for dehydroascorbic acid and the glucose analog, 2-deoxiglucose, in *Xenopus laevis* oocytes. All the mutants decreased the uptake of both substrates to less than 50%. The partial effect of the N317A mutant in transporting dehydroascorbic acid was associated with a 30% decrease in the V_max_ compared to the wildtype GLUT1. The results show that both substrates share the eight residues studied in GLUT1, albeit with a differential contribution of N317. Our work, combining docking with functional studies, marks the first to identify structural determinants of oxidized vitamin C’s transport via GLUT1.

## 1. Introduction

The family of facilitative glucose transporters (GLUTs) is composed of 14 members grouped into three classes according to their topology and substrate specificity: Class I (GLUT1-4), Class II (GLUT 5, 9, and 11), and Class III (GLUT 8, 10, and 12) [1]. GLUT1 was the first member of the family identified, namely in erythrocytes in 1977, while GLUT14, cloned in 2002, was the last [1,2]. Although the different members of the family have tissue-specific localization, GLUT1 is not only ubiquitous but also essential for glucose uptake in all of the body’s cells. The neurological alterations in GLUT1 deficiency syndrome (De Vivo disease), including infantile seizures and developmental delay, reveal the importance of GLUT1’s expression in the transport of glucose through the blood-brain barrier [3]. De Vivo disease is a hereditary pathology involving low glucose concentration in the cerebrospinal fluid associated with GLUT1’s impaired expression at the brain-blood barrier [4]. Otherwise, GLUT1’s overexpression in tumor cells reinforces its importance in the transport of glucose for energy supply [5].

GLUT1 also transports dehydroascorbic acid (DHA), the oxidized form of vitamin C, a soluble antioxidant with a plasma concentration between 50 and 100 μM [6]. Oxidative stress, an imbalance between oxidant and antioxidant agents, has been associated with processes such as inflammation and aging, and diseases such as cancer, obesity, diabetes, and atherosclerosis [7]. Vitamin C is acquired by cells primarily via the sodium ascorbate co-transporters, SVCT1 and 2, as ascorbic acid (AA), the reduced form of the vitamin [8]. However, under the local and transient oxidation of AA, cells can acquire DHA by a bystander process, as was confirmed in endothelial cells co-cultured with active neutrophils [9,10]. By contrast, erythrocytes, as cells that do not express SVCT transporters, acquire vitamin C only in its oxidized form via GLUT1 [11]. Because glucose is 100-fold more concentrated in the plasma than vitamin C, stomatin is necessary to increase the affinity of GLUT1 for DHA [12]. The uptake of DHA by GLUT1 in erythrocytes constitutes an evolutionary adaptation of organisms that cannot synthesize vitamin C, and in the case of De Vivo disease, a reduction of DHA’s uptake in erythrocytes has been reported [11,13,14]. Moreover, the presence of the GLUT1 transporter in the mitochondria and its increase in cancer cells have been associated with its capacity to transport DHA, contributing to the maintenance of a safe oxidative status in the mitochondria of cancer cells [15,16].

Along with its antioxidant capacity, vitamin C is also a cofactor of hydrolase enzymes related to the synthesis of collagen and neurotransmitters [17]. The capacity of GLUT10 to transport DHA is relevant to the synthesis of collagen in the arteries [18]. As a case in point, arterial tortuosity syndrome is a rare genetic disease characterized by the lengthening, distortion, and twisting of arteries where the synthesis of collagen and elastin is impaired due to GLUT10 mutations [19]. Other members of the GLUT family, including GLUT2, GLUT3, GLUT4, GLUT6, and GLUT12, also transport DHA, but their relevance for the physiology of vitamin C has not been explored [8].

The relationships between the structure and the function of hexose transporters in the GLUT family have been widely explored, especially in the case of GLUT1 [13,20,21,22,23,24,25,26]. This has been made possible by the obtention of the crystal structure of GLUT1, as well as the three-dimensional models based on the crystallized structures of several bacterial and fungi transporters [24]. The related transporters used have been lactose permease, glycerol-3-phosphate permease, fucose transporter, and xylose permease from *Escherichia coli* [21,25,26]. The 492 residues of GLUT1 have been shown to be arranged in 12 transmembrane helices (TMs) separated into two structurally overlapping domains with the amino and carboxyl termini facing the cytosol [1,24] (Figure 1). There are six amphipathic helices that form the transport channel and determine the substrate specificity (TM2, 4, 5, 7, 8, 10) and four helices facing the intracellular space (TM1-4) [1,2]. The principal signature motifs for glucose transport are PESPRY/FLL in a loop between TM6 and TM7, and QQLSGIN in TM7 [2,20]. The prolines and tryptophan in TM10 are relevant as well, as shown in Figure 1 [20].

Given the lack of data on the amino acid residues relevant for the transport of DHA via members of the GLUT family, in our study we aimed to identify the amino acid residues that are important for the acquisition of DHA by human GLUT1 by complementing the bioinformatics with functional assays. We hypothesized that DHA shares some important already-identified residues for glucose transport mediated by GLUT1.

## 2. Results

### 2.1. Docking Study of DHA in GLUT1

We performed a comparative analysis of the interaction of GLUT1 with DHA, AA, and several hexoses (i.e., glucose, glucose-6-phosphate [G-6-P], galactose, mannose, and glucosamine) in a docking restricted to a 5 Å grid located in the glucose binding cavity using the published crystallographic structure of GLUT1 (Figure 2A and Table 1). Considering the docking scores and the theoretical values of binding free energy (∆G_bind_), we observed that the most favorable interactions were for glucose, galactose, and glucosamine, and the least favorable for G-6-P and AA (Figure 2B,C and Table 1). When AA, as a lactone, captures two electrons, it generates DHA, which replaces the two adjacent hydroxyl groups with two carbonyl groups. In solution, DHA is hydrated and forms a second ring that fuses to the first to originate its bicyclic hemiacetal form. This last form of DHA presents a more favorable interaction than the non-hydrated form, DHA(bi), which presents a more favorable interaction than the non-hydrated form (Figure 2C,D and Table 1). Mannose and galactose are epimers that differ from glucose in the configuration of the chiral carbons, C2 and C4, respectively; while in glucosamine, the hydroxyl attached to the C2 of glucose is replaced by an amino group. DHA and its bicyclic form differ from the pyranose structure of the sugars carried by GLUT1, but also interact with that transporter (Figure 2D and Table 1).

Next, we performed a docking study with DHA(bi) and glucose in the complex with GLUT1, again centered in the 5 Å grid. Figure 3 shows the interaction between GLUT1 with glucose and DHA in one of the lower energy conformations. The residues F26, Q161, I164, Q282, Y292, W388, and W412 showed a high binding capacity for both substrates, whereas T30 and N317 only interacted with DHA (Figure 3A, residues in red). We also observed that other hydrogen bonds stabilized the interaction of GLUT1 with both substrates: T30, N288, and E300 with DHA(bi); and Q282 and Y292 with glucose. The docking simulation demonstrated that though the surface of GLUT1 involved in the interactions with DHA(bi) and glucose were similar and involved residues located in TM1, TM5, TM7, TM10, and TM11, DHA(bi) interacts differentially with eight residues and glucose with six (Table 2).

A more detailed analysis of all the docking assays revealed that glucose also formed a hydrogen bond with residues Q161, N288, G384, and W388, as well as with Y298 via a water molecule. Beyond that, residues I164 and F379 interact with the C6 hydroxymethyl group for glucose via van der Waals contacts (data not shown).

### 2.2. Functional Studies on DHA’s Transport via the Eight GLUT1 Mutants

We performed an alanine scanning for the eight residues in GLUT1: six were selected from our docking analysis; two had a high binding capacity for glucose and DHA (i.e., F26 and Y162); four interacted only with glucose (i.e., Q282, N317, W388, and Y412); and one interacted only with DHA (i.e., I164). Additionally, we included Q161, a residue that was important for glucose binding in the center of the transport pore according to the crystallographic structure published by Deng et al. [24]. For most of the mutants studied, decreases of similar magnitudes of the initial transport rate (V_o_) for DOG and DHA were observed, with a 5–80% decrease with respect to the wildtype GLUT1 (Figure 4). By contrast, the W388A mutant presented a lower transport of DHA (i.e., a 95% vs. 70% decrease, *p* < 0.05), whereas the N317A mutant showed higher levels (i.e., a 53% vs. 73% decrease, n.s.). These data reveal that glucose and DHA share most of the residues analyzed, with W388 and N317 being the differential functional residues.

### 2.3. Functional Characterization of the DHA Transport via the GLUT1 Mutant N317A

Because the mutant N317A presented a differential effect on the transport of DHA and DOG, we analyzed with more detail the effect of the mutation in the kinetics of the DHA substrate. We observed in the DHA saturation curve that the mutation produced an evident decrease in maximum velocity (V_max_) (Figure 5A,B). When the data were analyzed by the Eadie-Hofstee transformation, a Michaelis-Menten kinetic constant (K_m_) close to the wildtype GLUT1 was observed (i.e., 3.1 mM vs. 2.5 mM) along with a 30% decrease in the V_max_ (i.e., 60 vs. 88 pmoles/[oocyte × min]). Moreover, when we investigated the effect of the different hexoses (glucose, galactose, and glucosamine) in the DHA transport via GLUT1, we found a higher effect in the V_o_ in response to glucose and a similar response to galactose and glucosamine, observing more than a 75% decrease at 30–50 mM glucose (Figure 5C) and at 10–30 mM of galactose and glucosamine (Figure 5D).

## 3. Discussion

The results of our docking studies align with the functional data available: glucose, galactose, glucosamine, and DHA(bi) but not AA nor G-6-P show favorable interactions with GLUT1 [19]. The difference in the docking score between DOG and DHA(bi) suggests the existence of partly common structural determinants of the transport of both substrates, which also aligns with the functional data available because the K_m_ for DHA was half of that reported for DOG in GLUT1 (i.e., DHA 1.5 mM vs. DOG 2–4 mM) [6]. Similar results have been found for GLUT3 (i.e., DHA 2.0 mM vs. DOG 1.0 mM). However, for GLUT4 and GLUT2, the opposite is the case. The K_m_ for DHA is double the value for DOG (i.e., GLUT4 DHA 1.0 mM vs. DOG 5.0 mM, GLUT2 DHA 6 mM vs. DOG 12 mM) [8]. These data reinforce the idea that there are functional differences in the transport of both substrates and this is mediated by different isoforms of the GLUT family. The fructose transport in GLUT2 and GLUT5, for instance, has also been shown to be associated with the absence of the QLS sequence in TM7 [21]. Likewise, the conservation of residues has been found to be correlated with various kinetic properties conserved—for example, in GLUT1 (i.e., >97% between humans, mice, rats, rabbits, and pigs), correlates with a K_m_ value for DOG are between 1 and 2 mM along with the kinetic values for the reciprocal competitive inhibition between DHA and DOG (i.e., IC_50_ and inhibition constant) [1,6,13,23].

In accordance with these data, our docking analysis on the 5 Å grid of the transport pore revealed that of the 13 residues that interacted with DHA, eight did not interact with DOG. However, in the functional assays, all of them affected the transport of both substrates, although a partly different effect was observed between both substrates for W388 and N317. That difference could be due to the greater or lesser strengths of the H-bond interactions established or the grade of those interactions in the transport cycle of DOG and DHA. It is also plausible that the changes were due to the relative weak expression of the mutants at the plasma membrane level. Along those lines, the available data show that the Q161L, Q161N, N317C, and Y292F mutants do not change the expression of GLUT1 in the plasma membrane, but decrease DOG’s uptake in the oocytes, and the same have been observed for the 19 plausible mutants of W388 expressed in yeast and for the W412L mutant expressed in CHO cells [20,27,28,29]. Moreover, the transport dynamics revealed in molecular dynamic simulation studies are compatible with a homo-tetramer conformation, which functions as two dimers that alternately present a glucose-binding site on the external and internal faces of the transporter [24,28,29]. Recently, Galochkina et al. have also reported a strong coupling between the intracellular and extracellular domains of the transporter that allows for the adequate reorientation of the helices during the transport cycle [30]. Added to that, they found that glucose rotates as it is transported, which modifies the rotation of the alternative chains of different residues but maintains the same H-bonds [30]. Concerning the eight site-directed mutations analyzed in our study, all eight had already been studied in the context of DOG’s transport of GLUT1 [25,26,27,28,29]. For example, when Galochkina et al. modeled GLUT1 in the form of a holoenzyme, they found that the E380 side chain establishes a H-bond with glucose in the outward open conformation, whereas the residues T30, Q282, Q283, W388, and N288, did so in both conformations, preferably in the inward open conformation for T30 and the outward conformation for W388 and N288 [31]. At the same time, it has been proposed that the aromatic amino acids, F26 and W388, would prevent the exit of glucose through the transport pore [31,32,33]. It has additionally been reported that W412 forms a weak H-bond with glucose that allows its displacement through the transport pore, and that Y292 interacts with glucose via hydrophobic and hydrophilic interactions [21]. It is plausible that all the members of the GLUT family share the topology and transport cycle of GLUT1, and that the residues identified as relevant to DHA’s transport mediated by GLUT1 are also relevant among the other members of the family that also transport DHA because most of them are highly conserved (Figure 6).

The importance of GLUT1 for the physiology of vitamin C is particularly relevant in erythrocytes, which do not express SVCT1-2. In those cells, the association of GLUT1 with stomatin favors the transport of DHA over glucose transport even though the plasma concentration of vitamin C is less than 100 μM while that of glucose is 4–8 mM [33]. On that topic, Tu et al. have also revealed that GLUT1 is necessary to maintain the integrity in erythrocytes as well as the ascorbate plasma concentrations, both in vitro and in vivo, by mediating a transmembrane electron transfer system [34]. Thus, DHA’s transport by GLUT1 in erythrocytes has been proposed as an evolution-appropriate condition in species unable to synthetize vitamin C, including humans [35]. In other cell types, including astrocytes, melanoma cells, and endothelial cells, particularly those of the blood-retinal, blood-brain, and cerebrospinal fluid-blood barriers, DHA’s transport by GLUT1 is possible by the local and transient oxidation of AA and is relevant to maintain cerebral vitamin C levels [6,9,10]. Particularly in mitochondria and cancer cells, DHA’s transport via GLUT1 has been associated with antioxidant defense [15,16].

Our results reveal that DHA’s transport mediated by GLUT1 shares several residues already identified as important for glucose uptake by GLUT1, with W388 and N317 highlighted as residues having differential functional effects for both substrates. Our study marks the first to identify the structural determinants of DHA’s transport in GLUT1, namely by combining docking predictions with functional studies. Taken together, this reveals the importance of understanding how GLUT1 binds with and transports DHA, and encourages future research on GLUT1’s participation in vitamin C recycling, intracellular compartmentalization, and tissue distribution. Among those lines, it could be especially valuable to study the effects of De Vivo disease on cerebral vitamin C levels and their association with the neurological symptoms of the disease, as well as to explore the impaired transport of DHA in other chronic diseases.

## 4. Materials and Methods

### 4.1. Docking Simulation

For the in silico analysis of the interaction of DHA with GLUT1, we performed molecular docking of DHA (CID: 440667) and glucose (CID: 5793) using the GLUT1 crystallographic structure (PDB ID: 4PYP) published by Deng et al., which captures the transporter in a conformation partly open to the endofacial face [24]. The GLUT1 structure was energetically minimized in a Polak-Ribière conjugate gradient until reaching a convergence threshold of 0.05 using the software, MacroModel (Schrödinger, LLC, New York, NY, USA, 2020). Prior to the docking analysis, residues T45 and E329 were reincorporated by in silico mutagenesis using the software, Maestro (Schrödinger, LLC, New York, NY, USA, 2020). They had been mutated to reduce the conformational mobility of the transporter and facilitate its crystallization. We performed a comparative analysis of the interaction of GLUT1 with DHA, glucose, and other related molecules, such as G-6-P (CID: 5958), galactose (CID: 6036), and glucosamine (CID: 439213), by site-directed docking restricted to a 5 Å grid located in the binding cavity for glucose using Glide with an extra precision (XP) configuration, which included a post-docking minimization with ten poses per ligand (Schrödinger, LLC, New York, NY, USA, 2020). In the case of DHA, the DHA(bi) form (CID: 90659000) was also considered. The docking score for each interaction was obtained using Glide, and the theoretical ∆G_bind_ for each molecule with GLUT1 was calculated with the software, Prime (Schrödinger, LLC, New York, NY, USA, 2020), using the method of molecular mechanics-generalized Born surface area method (MM-GBSA) and the VSGB solvation model, and considering the OPLS4 force field [36] (Schrödinger, LLC, New York, NY, USA, 2020). All the bioinformatic figures were created using PyMol (DeLano Scientific, San Carlos, CA, USA, 2002).

### 4.2. GLUT1 Mutants 

The full-length GLUT1 cDNA cloned in pAGA was used to generate the different alanine mutants with the QuikChange site-directed mutagenesis kit (Stratagene, CA, USA), in accordance with the manufacturer’s instructions. An alanine screening of eight mutants was performed: F26A, Q161A, I164A, Q282A, Y292A, N317A, W388A, and W412A. Each clone was subjected to automated sequencing to verify the introduction of the desired mutation. For GLUT1 expression in *Xenopus laevis* oocytes, cRNA was synthesized in vitro using the mMessage mMachine kit (Ambion, Life Technology, Carlsbad, CA, USA).

### 4.3. Animal Procedure

Frogs were maintained in dechlorinated water at 18 °C in an inverted light-dark cycle and fed every two days. Before surgery, the animals were anesthetized by immersion in 0.03% benzocaine, and 40 ng of each cRNA were injected in defolliculated oocytes at stage V using a micromanipulator Nanojet II (Drummond Scientific, Bromall, PA, USA). Uptake assays were performed three days after cRNA injection [37]. The Institutional Ethics Committee approved the procedure, which conformed to the Guide of the Care and Use of Laboratory Animals of Chile’s National Council for Science and Technology Research (CONICYT, Chile).

### 4.4. Transport Assays

Uptake assays were performed as previously reported [37]. The oocytes were incubated in a transport buffer (i.e., 15 mM HEPES [pH 7.4], 135 mM NaCl, 5 mM KCl, 1.8 mM CaCl_2_, and 0.8 mM MgCl_2_). DHA uptake assays were performed in the transport buffer plus 1 μCi/mL of L-[1^4^C]-dehydroascorbic acid (i.e., specific activity 8.2 mCi/mmol, DuPont NEN, Boston, MA, USA) and cold non-radioactive DHA until reaching a final concentration of 0.5–5 mM. Beforehand, AA was oxidized with 1–10 U of AA oxidase (50 U/mg protein, Sigma-Aldrich, St. Louis, MO, USA) for 5 min at 37 °C. DOG uptake assays were similarly performed using 2 μCi/mL of [1,2-^3^H]-DOG complemented with non-radioactive DOG (specific activity 21.4 mCi/mmol, DuPont NEN, Boston, MA, USA). DOG, a non-metabolizable analog of glucose used in transport assays, is metabolized to 2-DOG-6-phosphate by hexokinase but cannot be further metabolized and is accumulated in cells [38]. The V_o_ was determined as the slope in independent time course assays. For DHA, the conditions were a concentration of 0.1 mM and a maximum time of 15 min. For DOG, the conditions used were a concentration of 0.5 mM and a maximum time of 10 min. These conditions ensured working under V_o_ without an evident saturation of the transport (data not shown). To obtain the kinetic parameters, a concentration of DHA or DOG between 0.2–5 mM was used. The assays were performed at 8 min. The kinetic parameters, K_m_ and V_max_, were determined by linear transformation of Eadie–Hofstee of the Michaelis–Menten equation. The effects of the hexoses (glucose, galactosamine, and glucosamine) on DHA’s transport were performed at 30–50 mM for glucose, and 10–30 mM for galactose and glucosamine at 8 min. The hexoses were added simultaneously with 0.1 mM of DHA. The uptake was finished with a cold-stopping solution (i.e., transport buffer plus 0.2 mM of HgCl_2_), which was washed twice and lysed in 300 μL of 10 mM Tris–HCl, 0.2% sodium dodecyl sulfate (pH 8.0). The incorporated radioactivity was determined by liquid scintillation in a counting system (Beckman Coulter, Brea, CA, USA) after the addition of EcoScint LS 6500 liquid (National Diagnostics, Atlanta, GA, USA).

### 4.5. Statistical Analyses

Data are presented as mean ± the standard deviation of a minimum of three assays performed in triplicates in transport assays. Statistical analyses were performed using the Tukey post hoc test on a one-way variance analysis, ANOVA, and *p* values of less than 0.05 (#) and less than 0.01 (*) were considered to indicate statistical significance.

## 5. Conclusions

The results of our study show that DHA’s transport via GLUT1 functionally involves the eight residues analyzed, of which W388 and N317 have minimal functional differences between both substrates. Our study thus marks the first to identify the structural determinants of DHA’s transport via GLUT1 by combining docking predictions with functional studies. Taken together, this reveals the importance of understanding how GLUT1 binds with and transports DHA, and encourages future research on its participation in vitamin C recycling, intracellular compartmentalization, and tissue distribution. Along those lines, it would be especially valuable to study the effects of De Vivo disease on vitamin C levels in the central nervous system and their association with the neurological symptoms of the disease, as well as to explore the impaired transport of DHA in other chronic diseases.

## Figures and Tables

**Figure 1 molecules-28-00521-f001:**
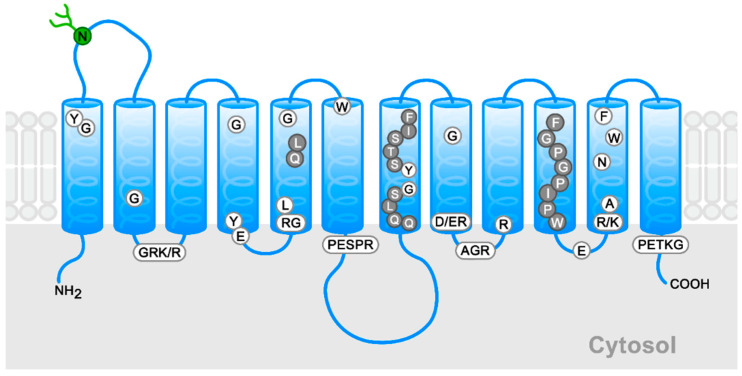
Schematic general model for members of the GLUT family. The conserved signature sequences appear in boxes, conserved residues in white circles, and high conserved residues in gray circles. The N-linked glycosylation site of Class I is indicated in green. The schema is based on the new GLUT1 crystal structure [24].

**Figure 2 molecules-28-00521-f002:**
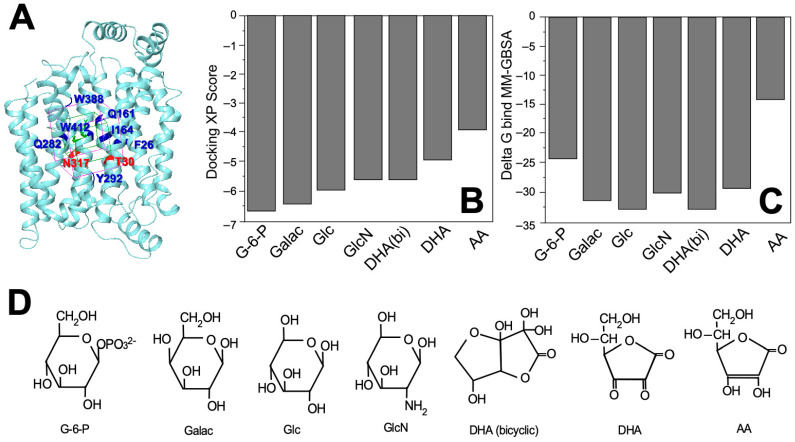
Interactions between GLUT1 and the different substrates as predicted by protein-ligand docking. (**A**) Position of the 5 Å grid interaction (purple box) used for docking simulations with the amino acids that are part of the sugar-binding sites indicated in blue and red; (**B**) Docking XP scores for the seven molecules capable of forming a stable complex with GLUT1 (i.e., glucose-6-phosphate, D-galactose, D-glucose, D-glucosamine, dehydroascorbic acid, bicyclic dehydroascorbic acid, and ascorbic acid); (**C**) Theoretical ∆G_bind_ values for the seven molecules analyzed with data expressed in kcal/mol and estimated using the MM-GBSA method; (**D**) Structure of the seven molecules analyzed in this study. G-6-P: glucose-6-phosphate; Galac: D-galactose; Glc: D-glucose; GlcN: D-glucosamine; DHA (bicyclic): dehydroascobic acid bicyclic form; DHA: dehydroascorbic acid; AA: ascorbic acid.

**Figure 3 molecules-28-00521-f003:**
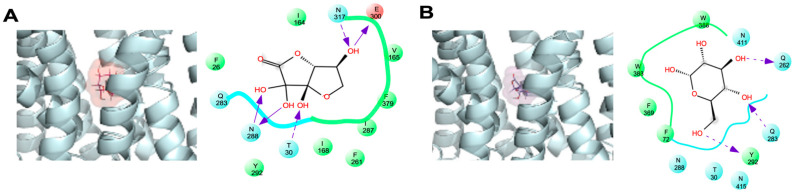
Comparative analysis of the DHA(bi) and glucose-binding mode of GLUT1 in the 5 Å grid. Representative complexes of GLUT1 with (**A**) DHA(bi) and (**B**) glucose are shown. The docking simulations were performed in a 5 Å grid of the GLUT1 junction pocket. Amino acid residues that interact with each molecule are indicated; the arrows indicate specific interactions between each substrate and the amino acid residues in GLUT1. Light dotted arrows indicate H-bond interactions with amino acid side chains. Negatively charged residues appear in red, hydrophobic residues in green, and uncharged polar residues in blue.

**Figure 4 molecules-28-00521-f004:**
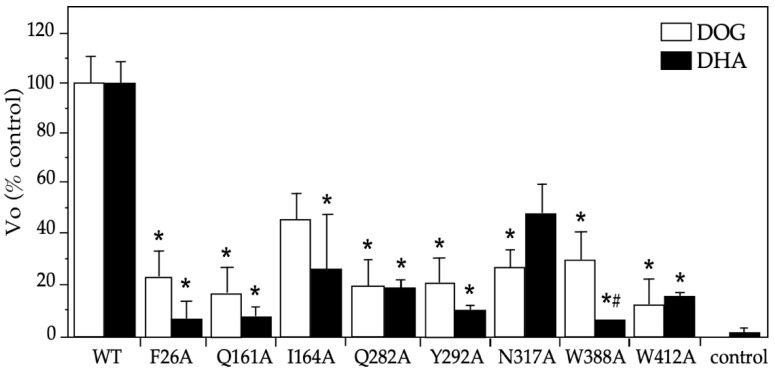
Initial rates for DOG and DHA transport mediated by GLUT1 mutants in *Xenopus laevis* oocytes. Uptake was carried out at 2 min intervals using DOG (0.5 mM) and DHA (0.1 mM), and the slopes of the time-curse plots were graphed. The maximum transport time was 10 min for DOG and 15 min for DHA. The *X. laevis* oocytes were injected with 40 ng of hGLUT1 cRNA or its mutants or if in the control, an equal volume of water. Three experiments were carried out in triplicates, counting four oocytes per sample. The data are expressed as mean ± SD with respect to the WT (6.30 ± 0.68 pmoles/oocyte × min for DOG and 6.30 ± 0.60 pmoles/oocyte × min). DHA: dehydroascorbic acid, DOG: 2-deoxyglucose, WT: wildtype GLUT1, Vo: initial rate. * *p* > 0.01 with respect to the control substrate and # *p* < 0.05 DHA transport with respect to DOG transport by the same mutant.

**Figure 5 molecules-28-00521-f005:**
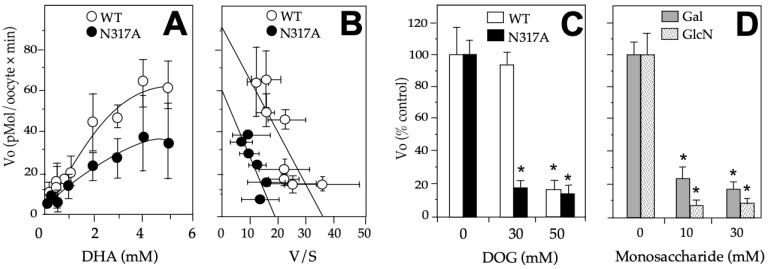
Effects of N317A on DHA transport on GLUT1. (**A**) DHA saturation curves of GLUT1 wildtype and N317A mutant; (**B**) Eadie-Hofstee transformation of the data shown in (**A**); (**C**) Effect of N317 in response to glucose. DHA’s transport in mutant N317A was performed in the presence of 30–50 mM glucose. (**D**) Effect of N317 in response to galactose and glucosamine. DHA’s transport in mutant N317A was performed in the presence of 10–30 mM galactose and galactosamine DHA. The data are expressed as mean ± SD of three experiments performed in triplicates. In (**C**,**D**), the results are presented as a percentage of the wildtype or mutant in the absence of the hexose explored, and 0.1 mM DHA was used. *Xenopus laevis* oocytes were injected with 40 ng of hGLUT1 or N317A mutant cRNA or if in the control, with equal volume of water. All the experiments were done at 8 min. For competition assays, hexoses were added simultaneously with DHA. DHA: dehydroascorbic acid, DOG: 2-deoxyglucose, Vo: initial rate, V: velocity, S: substrate, WT: wildtype GLUT1, Galac: galactose, GlcN: glucosamine. * *p* > 0.01.

**Figure 6 molecules-28-00521-f006:**
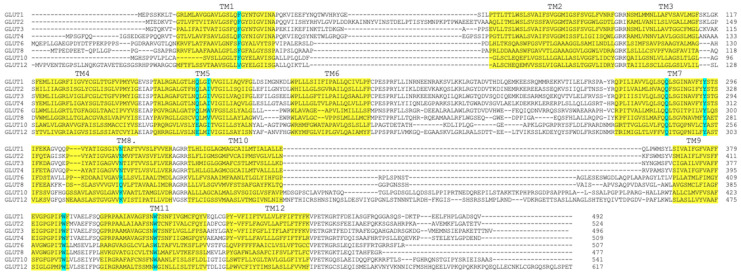
Amino acid sequence alignment of DHA transporters. Transmembrane domains are highlighted in yellow based on the new GLUT1 crystal structure [20,24]. The GLUT1 residues in our study and the equivalent residues among the other members of the GLUT family that transport DHA are marked in blue. Alignment was performed using Clustal 0 (1.2.4).

**Table 1 molecules-28-00521-t001:** Summary of the structural and energy parameters obtained from the protein-ligand docking predictions between GLUT1, hexoses, and DHA-related molecules.

Molecule	Docking Score	∆G_bind_ (kcal/mol)
Glucose-6-phoshpate	−6.620	−24.46
D-Galactose	−6.469	−31.58
D-Glucose	−5.959	−33.37
D-Glucosamine	−5.628	−30.53
Dehydroascorbic acid (bicyclic form)	−5.585	−33.12
Dehydroascorbic acid	−4.874	−29.99
Ascorbic acid	−3.885	−14.49

**Table 2 molecules-28-00521-t002:** Summary of the residues that interact with glucose and DHA(bi) in the 5 Å grid of GLUT1.

Substrate	Amino Acid Residue
DHA(bi) Glucose	F26, R30, **I164**, **V165**, **I168**, **F291**, Q283, **I287**, N288, Y292, **E300**, **N317**, **F379** F26, R30, **F72**, **Q282**, Q283, N288, Y292, **W388**, **N411**, **W412**, **N415**

Residues that interact only with the dehydroascorbic acid hemiacetal form [DHA(bi)] or glucose are bolded in red and black, respectively.

## Data Availability

The data presented in this study are available on request from the corresponding author.

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
