# Peer review of "Identification of Structural Determinants of the Transport of the Dehydroascorbic Acid Mediated by Glucose Transport GLUT1"

_molecules, 2023, doi:10.3390/molecules28020521_

Round 1

Reviewer 1 Report

The authors in this study attempt to screen for the amino acid residues involved in the binding of different substrate of GLUT1. The reviewer feels the manuscript makes no significant contribution to the understanding of the main goal of this manuscript as it has many serious flaws in the overall study design.

The introduction is poorly presented, and multiple sentences are not constructed well with no references for a factual statement. The authors needs to put in more effort to write this section.

For example: line 40-41, 47-51, 71-72 has no references cited.

Line 44, the word “multifunctionality” is not accurate here and the authors needs to work on lines 44-46.

Line 47-51, this paragraph start with no connection with the previous paragraph and the work “oxided is spelled wrong!

Line 59-61, the sentence formation needs major re-work.

Line 63-67, the wording “in terms of structure and function” need to be reworded. The authors need to reorganize the flow of information in the introduction, one suggestion is to start with  1. Focus on why study GLU1, 2. What is know about its substrates and binding pocket, 3. The overall topology (which can be better represented as a figure rather than a write-up) and sequence homology to know structures. 4. The working hypothesis and summary of what inference the study presents.

All figure legends are poorly worded.

Figure 1 legend needs major rework and in addition its mislabeled.

Figure 2 legend needs major rework. The reviewer would like to emphasize that a figure legend of “(A) Glucose interaction” (Line 122), is unacceptable to assume that readers would need to understand from just two words. The authors needs to put in more effort to word the figure legends to make sure it can be understood by the readers.

Figure 3, No justification is given as to why the concentration of DOG and DHA (0.5mM and 0.1mM) were used. Its not clear how the authors normalized the data and this detail should be given in the methods section. Following to the statement made in line 148-150, there is no clear connection from what we see in Figure 3 to the “justification” made by the authors. Figure 3 is very poorly presented, and the authors needs to work on it.

Why haven't the authors conducted a immunoblot analysis to test if the less significant (significant change claimed by the authors) change in Vmax/IC50 is due to change in protein expression levels?

The authors needs to understand the values they show are all within 2-fold change and is not considered significant to conclude that the change is significant. The method to conduct to see if a compound is a competitive substrate is inaccurate and the data points has large variability to conclude a credible data.

Author Response

REVIEWER 1

 We really appreciate your comments and your rigorous review of our manuscript. They will certainly improve the presentation of the context associated to our research and the relevance of our results.

Below, we present the changes made and the responses to your general evaluation of our manuscript.

1.- The new version of the manuscript was reviewed to another editing service. We really apology for the misspelling words.

2.- The introduction was re-written, to be more explicative

3.- Pertinent references were added where were necessary

4.- The aim was re-written centering in results obtained

5.- The methods were expanded and included more details about software and conditions used for bioinformatic analysis, and also for functional assays.

6.- Figure legends in the results section were corrected. Additionally, two tables were included, related to docking results and relevant residues in GLUT1.

7.- The conclusion was re-written, to be more explicative. A brief paragraph about future prospects of the study was also included.

Below, we answer your comments and suggestions, and also add that due to the comments of other reviewers, a list of abbreviation was additionally included to this new version of our manuscript

1.- The reviewer feels the manuscript makes no significant contribution to the understanding of the main goal of this manuscript and it has many serious flaws in the overall study design.

We do not agree with the statement that our manuscript has "serious flaws in the study design". Both types of assays used in our manuscript (bioinformatic and functional) followed similar approximations than those previously published by us and our collaborators (JBC, 2001; 276(48):44970-44975, Biophys J. ;87(5):2990-9. Biophys Res Commun. 2011 ;410(1):7-12; Nature. 1993 Jul 1;364(6432):79-82) and by anotheer researchers investigating functional and structural relationships in GLUTs  (J Biol Chem. 2012 Dec 14;287(51):42533-44, Am J Physiol Cell Physiol. 2015;308(10):C827-34). We believe that a wrong appreciation of the study design could have emerged due to our poor explanation of methods and lack of contextualization into the complex relationship between function and structure of glucose transporters. For that reason were-wrote several parts of the manuscript hoping that the changes clarify the rationale behind the study design and the experimental approximations used. Particularly, It is possible that the information related to topology and structure-function relationships in the GLUT family could result confusing and any help or suggestion to make it more compressive is welcomed. In this version of the manuscript we tried to make it more easy-reading, and believe that our work makes a relevant contribution to understand GLUT1 function as dehydroascorbic acid transporter. Moreover, not very often the published studies of GLUT transporters complements bioinformatic with functional assays.

2.- The introduction is poorly presented, and multiple sentences are not constructed well with no references for a factual statement. The authors need to put in more effort to write this section.

For example: line 40-41, 47-51, 71-72 has no references cited. Line 44, the word “multifunctionality” is not accurate here and the authors needs to work on lines 44-46. Line 47-51, this paragraph start with no connection with the previous paragraph and the work “oxided” is spelled wrong! Line 59-61, the sentence formation needs major re-work. Line 63-67, the wording “in terms of structure and function” need to be reworded. The authors need to reorganize the flow of information in the introduction, one suggestion is to start with 1. Focus on why study GLU1, 2. What is known about its substrates and binding pocket, 3. The overall topology (which can be better represented as a figure rather than a write-up) and sequence homology to know structures. 4. The working hypothesis and summary of what inference the study presents.

We followed your general advice regarding the introduction, modifying the logic behind the flow of information. Particularly, changes in lines 44-46, 47-51, 59-61, 63-67 were introduced to correct and improve the use of the language. Also new references were included in this new version (including the references suggested for lines 40-41, 47-51, 71-72). Moreover, we included an explanatory figure that shows the localization of conserved residues among GLUT family members, reducing the length of the descriptive text. Finally, we re-wrote the hypothesis emphasizing the relevance of our study.

3.- All figure legends are poorly worded. Figure 1 legend needs major rework and in addition its mislabeled. Figure 2 legend needs major rework. The reviewer would like to emphasize that a figure legend of “(A) Glucose interaction” (Line 122), is unacceptable to assume that readers would need to understand from just two words. The authors need to put in more effort to word the figure legends to make sure it can be understood by the readers. Figure 3, No justification is given as to why the concentration of DOG and DHA (0.5mM and 0.1mM) were used. Its not clear how the authors normalized the data and this detail should be given in the methods section. Following to the statement made in line 148-150, there is no clear connection from what we see in Figure 3 to the “justification” made by the authors. Figure 3 is very poorly presented, and the authors needs to work on it.

The legends were revised again, correcting the misspelling and expanding the explanation of each figure . Additionally, more information was included in methods section, regarding to software and molecules used in the simulation assays, concentration and time of transport assays. Also, data from control in transport assays were include in legends of figure 3 and 4 (new figures 5 and 6).

We presented the transport data in relation to control (wild-type GLUT1), because we are exploring changes introduced by the mutants, in DHA (and DOG) transport, which is more relevant that the absolute value. Anyway, we included the value of control DHA and DOG uptake in the legends as complementary data.

4.- Why haven't the authors conducted a immunoblot analysis to test if the less significant (significant change claimed by the authors) change in Vmax/IC50 is due to change in protein expression levels?

Your comment is very relevant. However, we could not perform western-blot in our assays when the assays were performed and now we do not have available the frogs. We explored alternatively, the expression of the mutants in HEK293 cells, merging the cDNA with GFP in another plasmid and did not find substantial changes in level of expression or localization. In the manuscript added a short paragraph abut previous studies published were western-blot of the same mutants were performed by others, and no evidence of changes in the expression of GLUT1 were found (. JBC. 1994, 12;269(32):20533-8.. JBC. 2004 12;279(11):10494-9. doi: 10.1074/jbc.M310786200, BBA  2009, 1788(5), 1051-1055.

5.-The authors need to understand the values they show are all within 2-fold change and is not considered significant to conclude that the change is significant.

We have to disagree with your comments, because we are responsible of our data analysis and the significance of the date were revealed by the statics test ANOVA was used. All the experiments were performed in triplicate and the test involve the multiple groups used in its analysis.

6.- The method to conduct to see if a compound is a competitive substrate is inaccurate and the data points has large variability to conclude a credible data.

We are aware that a linear transformation of the dose-response assay is necessary to formally demonstrate a competition and therefore, our data is not enough to conclude that such interaction is taking place between DOG and DHA. In our assay we did not intend to demonstrate competence between both substrates since that evidence has been previously published by others (XX). Instead. In our assays, we only explored possible changes in the effect of different hexoses in DHA't transport mediated by GLUT1 N317A mutant, respect to wild-type GLUT1, not including Ki values. In this new version the plot was changed to a bar graph with only selected concentration of glucose. A short explicative paragraph was included in methods about this misunderstanding.

We hope that new version which is enhanced by the observation of you and the other reviewers could be considered for you to publication in Molecules.

Sincerely,

Dr. Lorena Mardones

Associate Professor

Universidad Católica de la Santísima Concepción. Chile.

Reviewer 2 Report

In the paper entitled “Identification of structural determinants for dehydroascorbic acid in glucose transport GLUT1”, the interactions of dehydroascorbic acid in the glucose junction cavity (GLUT1) were investigated by molecular docking in comparison with those of sugars, and it was found that the interaction of the hemiacetal form of dehydroascorbic acid was inferior to those of sugars. Besides, the functional characterizations of the eight GLUT1 mutants were also investigated by transport assays and statistical analyses. This work could provide a theoretical basis for further researches of GLUT1. I recommend that this manuscript could be accepted for publication in your journal after minor revision.

1)      What were the software(s) and method(s) employed by the authors in the molecular docking studies of the manuscript? Please indicate these in the experimental section, and provide more details along with the raw data about the molecular docking studies, because it was important for the recurrence of the reported results.

2)      The research aim of this work was not clear, and the authors should clarify it in the abstract and introduction sections.

3)      In the docking simulation of this work, human glucose transporter GLUT1 (PDB ID: 4PYP) was used for the docking acceptor, but in the studies of GLUT1 mutants, frog (X. laevis) oocytes served as the experimental target. Therefore, how the authors regarded this difference and its affect on the relationship between the results of the docking simulation and animal experiment?

Author Response

REVIEWER 3

We really appreciate your comments of the manuscript.They will certainly improve the presentation and the relevance of our results.

Below we present the changes made and the responses to your comments.

1.- The new version of the manuscript was sent to another editing service.

2.- The introduction was re-written and re-organized from the general to the particular. Moreover, we included an explicative figure that shows the localization of conserved residues among GLUT family topology, reducing the descriptive text related to this issue. Finally, we reformulated the hypothesis to emphasize the relevance of our study.

3.- Pertinent references were added in modified part of introduction section

4.- The aim was re-written centering in results obtained

5.- The methods were expanded and included more details about software and conditions used for bioinformatic analysis, and the experimental conditions used in functional assays

6.- The figures in the results section and its legends were corrected. Additionally, two tables related to docking results and relevant residues were included,

7.- The conclusion was re-written, to be more explicative.

Regarding your suggestions, we included in this new version of our manuscript:

1.- What were the software(s) and method(s) employed by the authors in the molecular docking studies of the manuscript? Please indicate these in the experimental section, and provide more details along with the raw data about the molecular docking studies, because it was important for the recurrence of the reported results.
All the softwares used in docking assays were detailed in the new version of methods. Maestro software was used to carry out in silico mutagenesis and PyMOL software was used to create the figures added in the new version of the manuscript, complementing the softwares used in the original version of the
manuscript (MacroModel, Glide, Prime). A more detailed description was added for all steps required for docking simulations. Moreover, two tables of simulation results were included, one about free energy score (supplement data) and the other related to residues that interact with both substrates (DHA and glucoseresults section)

2.-The research aim of this work was not clear, and the authors should clarify it in the abstract and introduction sections.
Thank for your comment about this issue. Following your suggestion, the aim was re-written and include in abstract and introduction sections.

3.- In the docking simulation of this work, human glucose transporter GLUT1 (PDB ID: 4PYP) was used for the docking acceptor, but in the studies of GLUT1 mutants, frog (X. laevis) oocytes served as the experimental target. Therefore, how the authors regarded this difference and its affect on the relationship between the results of the docking simulation and animal experiment?

In the uptake assays, the full-length cDNA of GLUT1 from human origin was studied in Xenopus oocytes as an expression system. This model has been extensively used as expression model to perform functional assays of channels and transporter proteins, because it shows a low level of endogenous metabolite transport allowing to study the exogenous transporter with low interference of endogenous transporters-channels and high efficence in transcripction/translation of cRNA (i.e. PLoS One. 2011;6(7):e2190). Alternative expression systems used in GLUTs studies are mammalian cell lines including CHO, HEK293 or HeLa. However, they express a high level of endogenous GLUT1, and some of them also GLUT2, a low affinity transporter. Both transporters interfere in functional assays of DOG and/or DHA, hindering the study of GLUT1 transporter.

We hope that new version which is enhanced by the observation of you and the other reviewers could be considered for you to publication in Molecules.

Sincerely,

Dr. Lorena Mardones

Associate Professor

Universidad Católica de la Santísima Concepción. Chile.

Reviewer 3 Report

Identification of structural determinants for dehydroascorbic 2 acid in glucose transport GLUT1 is quite interesting and authors used combination of in-silico and experimental approaches to identify structural determinants.It could be interested for readers,

But here are few suggestion to improve this manuscript.

i)authors must write about future prospects of their study

ii) authors present their docking and simulation results in the table form.

iii)must provide a list of abbreviations

Author Response

REVIEWER 2

We really appreciate your comments of our manuscript. They will certainly improve the presentation and the relevance of our results

Below we present the changes made and the responses to your comments.

1.- The new version of the manuscript was sent to another editing service.

2.- The introduction was re-written and re-organized from the general to the particular. Moreover, we included an explicative figure that shows the localization of conserved residues among GLUT family topology, reducing the descriptive text related to this issue. Finally we re-formulated the hypothesis to emphasize the relevance of our study.

3.- Pertinent references were added in modified part of introduction section

4.- The aim was re-written centering in results obtained

5.- The methods were expanded and included more details about software and conditions used for bioinformatic analysis, and the experimental conditions used in functional assays

6.- The figures in the results section and its legends were modified. Additionally, two tables were included, related to docking results and relevant residues.

7.- The conclusion was re-written, to be more explicative.

In relation with your suggestions, we included in this new version of our manuscript:

1.- 2 table for docking results, related to score of free energy in simulation assays and a summary table with residues that interact with both substrates, describing the type interaction

2.- An abbreviation list.

3.- A paragraph about future prospects of the study at the end of the discussion.

We hope that new version which is enhanced by the observation of you and the other reviewers could be considered for you to publication in Molecules.

Sincerely,

Dr. Lorena Mardones

Associate Professor

Universidad Católica de la Santísima Concepción. Chile.

Round 2

Reviewer 1 Report

The authors have answered to reviewer's comments.

Minor comment:

Consider rewording line 61-62.